# NeoBabel: An Inclusive Multilingual Open Tower for Visual Generation

## Abstract

Text-to-image generation advancements have been predominantly English-centric, creating barriers for non-English speakers and perpetuating digital inequities. While existing systems rely on translation pipelines, these introduce semantic drift, computational overhead, and cultural misalignment. We introduce NeoBabel, a novel multilingual image generation framework that sets a new Pareto frontier in performance, efficiency and inclusivity, supporting six languages: *English, Chinese, Dutch, French, Hindi*, and *Persian*. The model is trained using a combination of large-scale multilingual pretraining and high-resolution instruction tuning. To evaluate its capabilities, we expand two English-only benchmarks to multilingual equivalents: m-GenEval and m-DPG. NeoBabel achieves state-of-the-art multilingual performance while retaining strong English capability. Notably, NeoBabel matches or exceeds English-only models while being 2–4× smaller. We release an open toolkit, including all code, model checkpoints, a curated dataset of 124M multilingual text-image pairs, and standardized multilingual evaluation protocols, to advance inclusive AI research.

## 1 Introduction

Recent advances in diffusion models and large-scale vision-language pretraining have revolutionized text-to-image generation, enabling the creation of high-quality images from natural language descriptions (Rombach et al., 2022; Peebles & Xie, 2023; Bao et al., 2023; Chen et al., 2024a; Xie et al., 2023; Wu et al., 2023a; Lipman et al., 2022; Xie et al., 2025a; Qin et al., 2025; Zhang et al., 2023; Seawead et al., 2025). Despite these remarkable capabilities, existing methods suffer from a critical limitation: an overwhelming reliance on English as the primary—and often exclusive—input language (Ramesh et al., 2022; Xie et al., 2025b; Chameleon Team, 2024). This monolingual bias creates substantial barriers for the billions of users who communicate in other languages, fundamentally restricting global access to state-of-the-art generative AI technologies (Bassignana et al., 2025; Peppin et al., 2025). The consequences of this linguistic limitation extend far beyond mere inconvenience. As text-to-image systems become integral to education, creative industries, art, and journalism, the lack of native multilingual support perpetuates existing digital divides and cultural inequities (Liu et al., 2023; Rege et al., 2025). Non-English speakers are forced to navigate through translation layers that not only introduce friction but also risk losing the nuanced meanings and cultural contexts that make their creative expressions unique (Kannen et al., 2024; Friedrich et al., 2024). Building truly multilingual models, like we do in this paper, is therefore not merely a technical challenge but an ethical imperative, one that ensures equitable access to generative AI while preserving linguistic diversity and cultural authenticity in the digital age.

Existing approaches to multilingual image generation typically employ a translation-first strategy, converting non-English prompts to English before processing. While this appears pragmatic, it introduces a cascade of problems that fundamentally compromise the user experience (Kreutzer et al., 2025; Li et al., 2025b; Bafna et al., 2025). The computational overhead of chaining translation and generation models effectively doubles inference time, creating prohibitive delays for real-time applications, thereby further disadvantaging non-English speakers. Most critically, this approach suffers from semantic drift—the systematic loss of culturally specific meanings and linguistic subtleties (Cohn-Gordon & Goodman, 2019; Vanmassenhove et al., 2019; Beinborn & Choenni, 2020). For instance consider the Dutch term "*gezellig*" which encompasses a complex blend of coziness, conviviality, and belonging and has no direct English equivalent. When forced through translation, such rich cultural concepts are inevitably flattened or distorted, resulting in generated images that

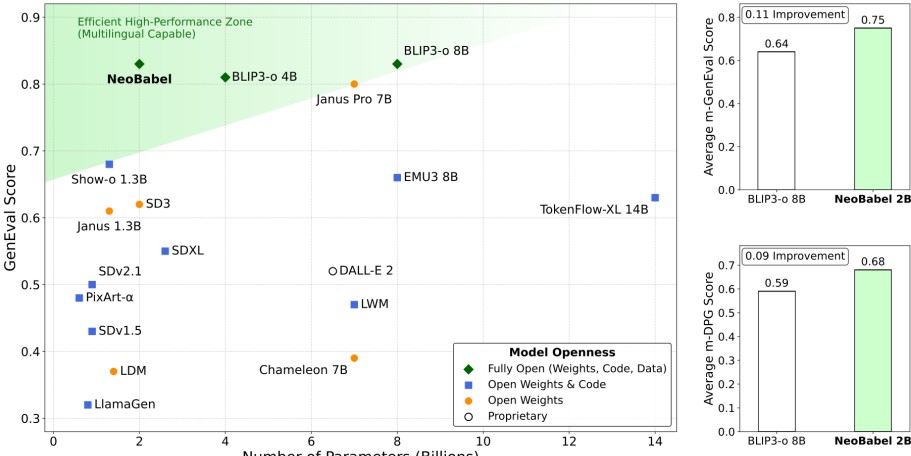

Figure 1: **NeoBabel establishes a new Pareto frontier in multilingual image generation performance, efficiency, and inclusivity.** Left: GenEval English-only scores show that NEOBABEL matches state-of-the-art models despite being 2–4× smaller. Right: On our multilingual benchmark extensions, m-GenEval and m-DPG, NEOBABEL outperforms the second-best model, demonstrating strong multilingual generalization. NEOBABEL is fully open (weights, code, data) and supports six languages with consistent cross-lingual performance.

fail to capture the intended meaning. The fundamental issue lies deeper than mere translation accuracy (Wein & Schneider, 2023; Singh et al., 2024; Salazar et al., 2025). We bypass this limitation by leveraging the image directly. We first caption an image in a target language (English) and then translate this English caption into other languages. The subsequent translations are image-grounded, as they are based on a description generated from the visual input itself.

This paper introduces NEOBABEL, a novel multilingual image generation framework that represents the first scalable solution for direct text-to-image synthesis across six languages: *English, Chinese, Dutch, French, Hindi*, and *Persian*. It achieves language-agnostic understanding without requiring translation, while maintaining performance parity that matches or exceeds English-only models across all languages (Figure 1). The NEOBABEL architecture delivers operational efficiency, processing multilingual prompts 2.8x faster than 'translation-then-generation' pipelines while using 59% less memory which is critical for real-world deployment scenarios (Section 2). To train the model, we introduce a data curation pipeline that prepares 124M multilingual image-text pairs (Section 3) for both pretraining and instruction tuning through progressive training stages (Section 4). We further introduce the first standardized framework for evaluating multilingual image generation, addressing critical gaps in existing benchmarks (Section 5). Our protocol includes: (1) extended versions of GenEval (Ghosh et al., 2023) and DPG-Bench (Hu et al., 2024), referred to as m-GenEval and m-DPG, across six languages, enabling direct comparison between native multilingual and translation-based approaches (Section 6). Finally, we release a comprehensive research toolkit comprising NEOBABEL model checkpoints trained on six languages, a systematically curated dataset of 124M multilingual text-image pairs with quality-controlled translations, and a complete reproducibility package including training scripts, hyperparameter configurations, and standardized evaluation protocols.

## 2 NEOBABEL ARCHITECTURE

Our architecture's core components, a multilingual tokenizer and transformer backbone, are optimized for efficient, scalable cross-lingual image generation, supporting seamless processing across diverse languages and image types. Figure 2 provides an overview of the NEOBABEL architecture.

**Tokenizers.** For textual input, we adopt the multilingual tokenizer of Gemma-2 (Gemma Team et al., 2024) without any modifications. This approach maintains compatibility with multilingual inputs while utilizing proven tokenization methods from language modeling. For image input, we leverage the MAGVIT-v2 quantizer (Yu et al., 2023) retrained by Show-o (Xie et al., 2025b) on 25 million images. This lookup-free quantizer learns a discrete codebook of size $K=8,192$ and encodes

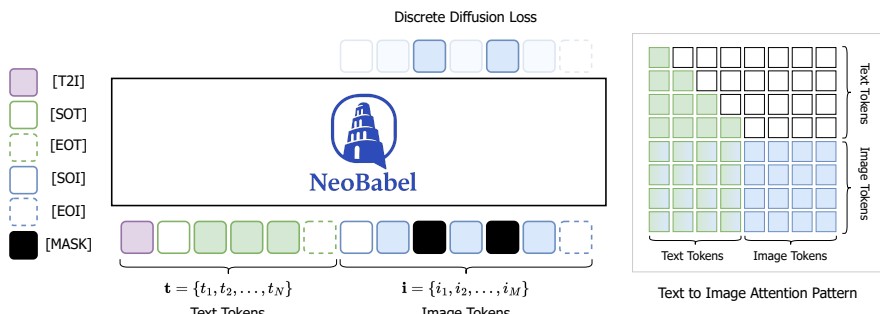

Figure 2: **NeoBabel architecture.** Regardless of modality, all input data is first tokenized and embedded into a unified input sequence. NEOBABEL then applies causal attention to text tokens and full attention within a discrete denoising diffusion framework for image tokens, ultimately generating the desired image.

$256 \times 256$ resolution images into $16 \times 16$ grids of discrete tokens. The quantization approach supports efficient downstream training and generation while preserving fine-grained visual details.

**Transformer Backbone.** As we build upon the pretrained multilingual large language model (LLM) Gemma-2 (Gemma Team et al., 2024), we maintain its overall transformer architecture, while introducing two key modifications: (1) integration of a unified multimodal embedding space, and (2) modality-aware attention patterns for flexible generation. Additionally, we apply qk-norm (Henry et al., 2020) to each attention layer to enhance training stability and convergence.

*Unified Multimodal Embedding and Prompt Design.* To enable seamless multimodal learning, we extend the LLM's embedding table with 8,192 new learnable embeddings for discrete image tokens, allowing the model to process image inputs natively without architectural changes. Both text and image tokens are embedded in a shared space, enabling the model to learn cross-modal compositionality and semantic alignment. We represent all tasks including text-to-image generation as unified autoregressive sequences. Given a tokenized image-text pair, text and image tokens are concatenated into a single sequence. Special tokens such as [T2I], [SOT], [EOT], [SOI], and [EOI] explicitly mark task type and modality boundaries, enabling the model to disambiguate different modalities and tasks through prompting alone. This design simplifies the training pipeline by removing the need for modality-specific components or task-specific heads, allowing for flexible, scalable, and unified multimodal generation.

*Modality-Aware Attention.* To accommodate differing structural needs of text and image modalities, we employ a hybrid attention mechanism. Text tokens are modeled with causal attention to preserve autoregressive language modeling capabilities. Image tokens are modeled using full bidirectional attention, allowing rich interactions critical for high-fidelity image synthesis. When both modalities are present, attention masks are dynamically configured so that image tokens can fully attend to text tokens and preceding image tokens, enabling coherent, contextually grounded generation.

**Training Objective.** The model is trained on sequences composed of both textual and visual tokens, where text tokens act as a prefix and visual tokens form the postfix. We do not apply any learning objective to the text tokens; the loss is computed solely over the visual tokens. Let $\mathbf{t} = \{t_1, t_2, \ldots, t_N\}$ denote the text tokens and $\mathbf{i} = \{i_1, i_2, \ldots, i_M\}$ denote the image tokens, forming a full input sequence $[\mathbf{t}; \mathbf{i}]$. During training, we randomly select a subset $\mathcal{J} \subset \{1, \ldots, M\}$ of image token indices to be masked. The corresponding masked sequence is denoted by $\mathbf{i}_*$, where $i_j$ is replaced with a special [MASK] token for all $j \in \mathcal{J}$. The model is trained to reconstruct the original visual tokens at the masked positions by conditioning on the full input sequence of text tokens and (partially masked) image tokens. The objective is defined as: $\mathcal{L} = \sum_{j \in \mathcal{J}} \log p_\theta(i_j \mid \mathbf{t}, \mathbf{i}_*)$, where $p_\theta(\cdot)$ is the model's predicted distribution over image codebook entries, parameterized by $\theta$. The loss is only applied to the masked image tokens in $\mathcal{J}$. We randomly mask a fixed ratio of visual tokens within each training sample (Xie et al., 2025b). To further improve generation controllability, we incorporate classifier-free guidance (Ho & Salimans, 2022) by replacing the conditioning text with a null string with some probability during training.

Table 1: **NeoBabel multilingual datasets**, detailing their English-only data source, image origin, caption format, and size. Our multilingual expansion covers model-generated recaptioning, translation into multiple languages, or both and increase the total size from 39M to 124M image–caption/label pairs. All modified datasets are prefixed with m- to denote their expanded form.

| Original English-Only Dataset | | | | NEOBABEL Multilingual Expansion | | |
|---|---|---|---|---|---|---|
| Dataset | Image Source | Caption Source | Size | Recaptioning | Translation | New Size |
| ImageNet 1K | Web | Class labels | 1M | – | ✓ | 6M |
| CC12M | Web | Alt-text (noisy) | 12M | ✓ | – | 12M |
| SA-1B | Photography | LLaVA | 10M | ✓ | – | 10M |
| LAION-Aesthetic | Web | Alt-text (noisy) | 12M | ✓ | ✓ | 72M |
| JourneyDB | Synthetic | GPT-3.5 | 4M | ✓ | ✓ | 24M |
| BLIP3-o Instruct | Web + Synthetic | GPT-4o / human | 60K | – | ✓ | 360K |
| | | | 39M | | | 124M |

## 3 NEOBABEL MULTILINGUAL DATASETS

**Data curation pipeline.** Multilingual multimodal data remains scarce, especially compared to the abundance of English-centric resources. This imbalance poses a significant barrier to training and evaluating models that can understand grounded language across diverse linguistic contexts. To address this gap, we curate and augment several multilingual datasets by translating and recaptioning existing image-caption pairs into six target languages: *English*, *Chinese*[1], *Dutch*, *French*, *Hindi*, and *Persian*. We summarize the datasets curated in Table 1. At the core of our approach is a multilingual captioning pipeline designed to ensure both semantic richness and linguistic diversity. We begin by generating a detailed English caption for each image using InternVL (Chen et al., 2024c), prompted with a simple instruction: "Describe this image in detail in English." This step guarantees comprehensive coverage of the visual content. To preserve quality and consistency across languages, we apply length filtering, language validation, visual-text mismatch and toxicity/NSFW filterings, with full details provided in Appendix A.5. Once high-quality English captions are obtained, we translate them into five target languages using the NLLB model (Costa-Jussà et al., 2022) for the pretraining datasets, and the Gemini Experimental model (gemini-2.0-flash-lite) for the instruction tuning datasets. Ultimately, this step plays a central role in constructing a high-quality, language-balanced multimodal resource, essential for more inclusive and globally-relevant vision-language models.

**NEOBABEL pretraining data.** We use a diverse collection of image-text datasets to build strong multilingual visual-language alignment combining real-world and synthetic image sources. While the images are drawn from established, high-quality datasets, the accompanying captions have been significantly enriched through our recaptioning and multilingual translation pipeline resulting in a more diverse, detailed, and valuable resource. The original English class labels are translated into five additional languages to obtain a total of six target languages, forming multilingual textual prompts for class-conditional image generation, denoted as m-ImageNet-1K. We further incorporate m-SA-1B and m-CC12M, consisting of 22 million English image-caption pairs (Kirillov et al., 2023; Changpinyo et al., 2021), which provide rich natural descriptions and enhance visual diversity; their texts are refined through our recaptioning pipeline. In addition, m-LAION-Aesthetic, a 12M subset of the LAION dataset(Clure, 2023), is enhanced and translated, yielding approximately 72 million image-caption pairs across six languages. Finally, m-JourneyDB, a synthetic dataset of 4 million high-quality Midjourney-generated images (Sun et al., 2023a), is processed with the same recaptioning and translation pipeline, resulting in 24 million multilingual pairs. Combining all sources, the final pretraining dataset contains approximately 124 million image-text pairs spanning six languages, covering diverse domains and visual aesthetics.

**NEOBABEL instruction tuning data.** We describe the datasets and mixing strategies used for instruction tuning. This phase reuses two datasets introduced earlier and adds a smaller but higher-quality dataset focused on multimodal supervision. Specifically, we continue to use m-LAION-Aesthetic and m-JourneyDB, as extended in the pretraining stage, and introduce m-BLIP3o-Instruct, an instruction-focused dataset from Chen et al. (2025a) containing multimodal instruction samples translated into six languages for multilingual training. All images are resized to $512 \times 512$. While

---

[1]Throughout this work 'Chinese' refers to Simplified Chinese.

the images are drawn from established, high-quality sources, most accompanying texts have been enriched or rewritten, resulting in a more valuable and linguistically diverse dataset.

# 4 NEOBABEL TRAINING STAGES: LEARNING PROGRESSION

NEOBABEL is trained using a staged learning framework consisting of three progressive pretraining stages followed by two instruction tuning stages. Training details are provided in Appendix A.2.

**Progressive pretraining.** Our pretraining progressively scales from basic visual understanding to advanced multilingual image generation. In Stage 1 (Pixel Dependency Learning), the model learns foundational visual representations using m-ImageNet-1K, where class-conditional generation is guided by translated class labels to capture pixel-level dependencies and build robust image token embeddings. Stage 2 (Scaling Alignment with Large-Scale Multilingual Data) fine-tunes the model on 22 million English-only image-caption pairs from m-SA-1B and m-CC12M, together with 72 million translated samples from m-LAION-Aesthetic, strengthening natural image-text alignment and expanding multilingual capabilities through broad cross-lingual exposure. Finally, Stage 3 (Refined Multilingual Pretraining) trains on 96 million multilingual pairs from m-LAION-Aesthetic and m-JourneyDB, balancing high-quality real-world aesthetic data with diverse synthetic images to enhance generalization across languages, domains, and modalities.

**Progressive instruction tuning.** In this phase, the model focuses on explicit task-guided adaptation, refining its ability to interpret and execute complex, multilingual instructions through our curated datasets and progressive exposure to prompt-driven generation in two stages. Stage 1 (Initial Multilingual Instruction Alignment) trains the model with a diverse mixture of m-LAION-Aesthetic, m-JourneyDB, and m-BLIP3o-Instruct using mixing weights $\alpha_1$, $\alpha_2$, and $\alpha_3$ such that $\alpha_1 + \alpha_2 + \alpha_3 = 100$. A higher $\alpha_1$ and moderate $\alpha_2$ prioritize real-world and aesthetic content, while a smaller $\alpha_3$ introduces early exposure to instruction-rich samples, helping the model learn cross-lingual, cross-modal grounding without being overwhelmed by complex prompts. In Stage 2 (Instruction Refinement), the mixing weights are shifted to emphasize instruction-rich and synthetic supervision by increasing $\alpha_2$ and $\alpha_3$ and reducing $\alpha_1$, enabling the model to refine its multilingual instruction-following abilities through complex prompts and high-quality synthetic images. This curriculum-style adjustment increases semantic richness and improves generalization to both benchmark instruction tasks and open-ended generation scenarios.

# 5 MULTILINGUAL EVALUATION OF IMAGE GENERATION

Existing image generation benchmarks are mostly English-centric, failing to capture cross-lingual performance. We introduce a multilingual evaluation suite that extends established (English-only) benchmarks to cover six diverse languages. We assess the image generation capabilities of NEO-BABEL using two complementary benchmarks: GenEval (Ghosh et al., 2023) and DPG-Bench (Hu et al., 2024). GenEval offers a structured evaluation of prompt-to-image alignment across six compositional dimensions: *single object*, *two objects*, *counting*, *colors*, *position*, and *color attribute*. In contrast, DPG-Bench targets general-purpose generation with open-ended, diverse prompts that test broader semantic understanding. However, both benchmarks are English-only and fail to capture multilingual generative performance. As part of our multilingual evaluation suite, we introduce **m-GenEval** and **m-DPG**, multilingual extensions of the original benchmarks. All prompts are translated into five additional languages: *Chinese*, *Dutch*, *French*, *Hindi*, and *Persian*, using the Gemini Experimental model, followed by human verification and manual corrections to ensure linguistic correctness. We also publicly release m-GenEval and m-DPG to promote inclusive and realistic evaluation of multilingual text-to-image models and support broader community adoption.

# 6 RESULTS AND DISCUSSIONS

## 6.1 MULTILINGUAL IMAGE GENERATION PERFORMANCE

**m-GenEval Comparison.** We evaluate NEOBABEL on the English prompts of the m-GenEval benchmark, with results reported in Table 2. The comparison includes both generative models (G), which focus solely on text-to-image generation, and unified models (U&G), which also support image understanding tasks such as captioning and visual question answering. Despite having only 2B parameters, NEOBABEL outperforms or matches best-performing unified models such as Janus-Pro 7B (0.80) and BLIP3-o 8B (0.83), which are larger in terms of parameters. It also surpasses SD3

Table 2: **English-only GenEval benchmark comparison.** NEOBABEL achieves the highest overall score, outperforming larger models on tasks requiring compositional reasoning and fine-grained prompt-image alignment. Symbol legend: ⊕ denotes multilingual generation capability, with ✓ indicates a full multilingual capability, ∘ represents partial multilingual capability (i.e. bilingual or multilingual to a limited extent), and × denotes monolingual models.

| Method | ⊕ | Type | Params. | Single Object | Two Object | Counting | Colors | Position | Color Attribute | Overall |
|---|---|---|---|---|---|---|---|---|---|---|
| LlamaGen | × | G | 0.8B | 0.71 | 0.34 | 0.21 | 0.58 | 0.07 | 0.04 | 0.32 |
| PixArt-alpha | × | G | 0.6B | 0.98 | 0.50 | 0.44 | 0.80 | 0.08 | 0.07 | 0.48 |
| SDXL | × | G | 2.6B | 0.98 | 0.74 | 0.39 | 0.85 | 0.15 | 0.23 | 0.55 |
| SD3 | × | G | 2B | 0.98 | 0.74 | 0.63 | 0.67 | 0.34 | 0.36 | 0.62 |
| Chameleon | × | U&G | 7B | - | - | - | - | - | - | 0.39 |
| LWM | ∘ | U&G | 7B | 0.93 | 0.41 | 0.46 | 0.79 | 0.09 | 0.15 | 0.47 |
| TokenFlow | ∘ | U&G | 14B | - | - | - | - | - | - | 0.63 |
| EMU3 | ∘ | U&G | 8B | - | - | - | - | - | - | 0.66 |
| Show-o | × | U&G | 1.3B | 0.98 | 0.80 | **0.66** | 0.84 | 0.31 | 0.50 | 0.68 |
| Janus-Pro | ∘ | U&G | 7B | - | - | - | - | - | - | 0.80 |
| BLIP3-o | ∘ | U&G | 4B | - | - | - | - | - | - | 0.81 |
| BLIP3-o | ∘ | U&G | 8B | - | - | - | - | - | - | **0.83** |
| NEOBABEL | ✓ | G | 2B | **1.00** | **0.91** | 0.62 | **0.91** | **0.81** | **0.77** | 0.83 |

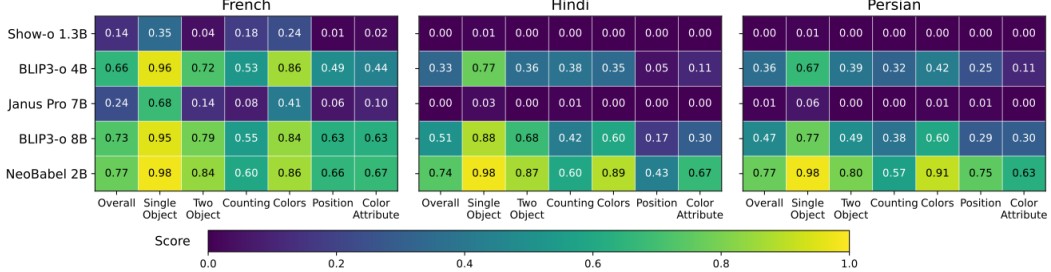

Figure 3: **m-GenEval benchmark comparison.** Models such as Janus Pro and BLIP3-o drop sharply in non-English settings, while NEOBABEL delivers consistent performance across languages (see Figure 10 in the appendix for all six languages).

2B (0.62), a leading model in the generative category, achieving the highest overall score of 0.83. This performance reflects strong fine-grained and compositional prompt-image alignment particularly in challenging subcategories like color attributes and positional grounding. In Figure 3, we report results on a representative subset of three languages, where NEOBABEL surpasses baselines in medium-resource French (0.04) and shows large gains in low-resource Hindi and Persian (up to 0.3). The full evaluation across six languages, provided in Figure 10 in the appendix, further reveals only a small gap in high-resource Chinese (0.03) and a moderate one in Dutch (0.06), highlighting the general trend that the advantage of NEOBABEL grows as resource availability decreases.

**m-DPG Comparison.** We evaluate NEOBABEL on m-DPG (Table 3), which measures semantic accuracy, detail, and coherence. With only 2B parameters, NEOBABEL provides moderate performance on English (0.75) compared to larger models i.e. BLIP3-o (8B) and Janus Pro (7B), while outperforming them in other languages. A consistent trend emerges: in medium-resource languages, the performance gap widens (e.g., +0.10 in Dutch, +0.09 in French), and in low-resource settings it broadens further (+0.13 in Hindi, +0.12 in Persian). This mirrors the pattern observed in m-GenEval, further underscoring the robustness of NEOBABEL in multilingual, low-resource scenarios.

## 6.2 QUALITATIVE EVALUATION

**Multilingual text-to-image generation.** We present qualitative results from NEOBABEL across diverse prompt categories, including compositional scenes, abstract concepts, and multilingual instructions. Examples are in Figure 4, with more in Figures 12 and 13 (appendix). As observed, objects, layouts, and attributes are preserved across languages. The results show that NEOBABEL consistently produces semantically aligned and visually coherent images.

Table 3: **m-DPG benchmark comparison.** Despite its small parameter count, NEOBABEL achieves competitive results in English and consistently outperforms all baselines across five non-English languages, demonstrating strong cross-lingual prompt understanding and image generation.

| Model | Params. | English | Chinese | Dutch | French | Hindi | Persian | Overall |
|---|---|---|---|---|---|---|---|---|
| Show-o | 1.3B | 0.67 | 0.10 | 0.22 | 0.32 | 0.04 | 0.04 | 0.23 |
| Janus Pro | 7B | **0.84** | 0.50 | 0.61 | 0.68 | 0.12 | 0.12 | 0.47 |
| BLIP3-o | 4B | 0.79 | 0.60 | 0.58 | 0.59 | 0.47 | 0.49 | 0.58 |
| BLIP3-o | 8B | 0.80 | 0.56 | 0.59 | 0.61 | 0.50 | 0.53 | 0.59 |
| NEOBABEL | 2B | 0.75 | **0.70** | **0.69** | **0.70** | **0.63** | **0.65** | **0.68** |

English: The image depicts a dynamic and vibrant scene featuring an abstract, three-dimensional cube suspended against a dark background with subtle red hues at the bottom edge. The cube is composed of fluid-like material that appears to be melting into various colors such as blue, pink, orange, yellow, and hints of purple. The liquid forms swirls around its edges like waves crashing over it, creating a sense of motion and energy within the composition. Bubbles are scattered throughout both inside and outside the cube's surface, adding texture and depth to the visual effect. The top part of the cube has more intense shades of blue while transitioning smoothly towards warmer tones on one side, giving off a fiery appearance where bright pinks and yellows dominate. This contrast creates a striking interplay between cool and warm colors across different sections of the object. Light reflections highlight some parts making them appear glossy and wet, enhancing the illusion of movement through light refraction effects visible along the curves formed by the flowing substance.

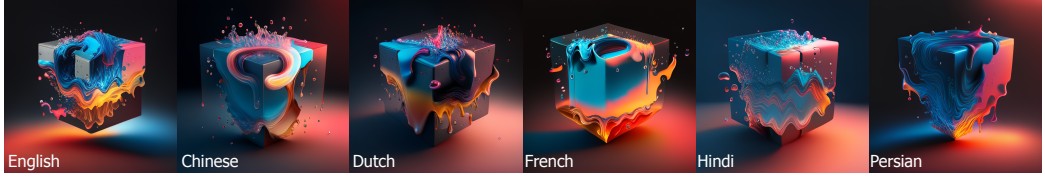

Chinese: 这张图描绘了一只迷人的动画猫角色，拥有大大的绿色眼睛和富有表现力的面孔，展现出自信与好奇心。猫咪戴着一顶细致的棕色帽子，帽侧装饰有螺丝和别针等配件。它身穿精致的服装，包括一件时尚的马甲，里面似乎是一件浅色衬衫。其毛发图案在耳朵和后腿周围有橙色斑点，赋予它独特的外观。背景为白色，突出了明亮复杂的颜色、衣服的细节和角色面部的表现。尾巴轻轻地在身后弯曲。整体上，这个设计很好地传达了大胆和好奇的感觉。

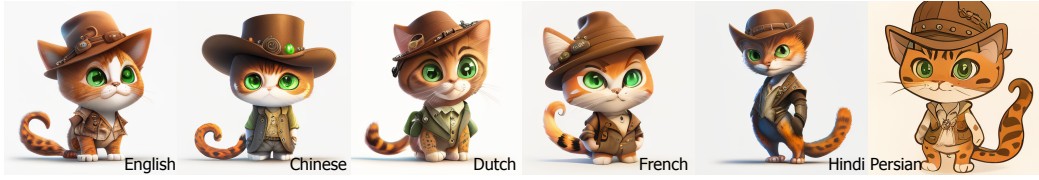

Figure 4: **Qualitative evaluation of NEOBABEL.** Each row is based on a single concept expressed in six different languages. We show only one of the prompts (in one language) and present six images generated from its translated prompts in the other five languages. Across all languages, NEOBABEL delivers semantically accurate and visually cohesive outputs with reliable consistency.

**Multilingual image inpainting and extrapolation.** NEOBABEL supports text-guided image inpainting and extrapolation across languages without additional fine-tuning. This enables collaborative applications, such as a multilingual visual canvas where users contribute prompts in their native languages to co-create expressive scenes. As shown in Figures 14 and 15, NEOBABEL can modify or extend an input image based on prompts in different languages, producing images that remain semantically faithful and visually consistent with the adjacent visual content, highlighting its potential for interactive and multilingual visual editing.

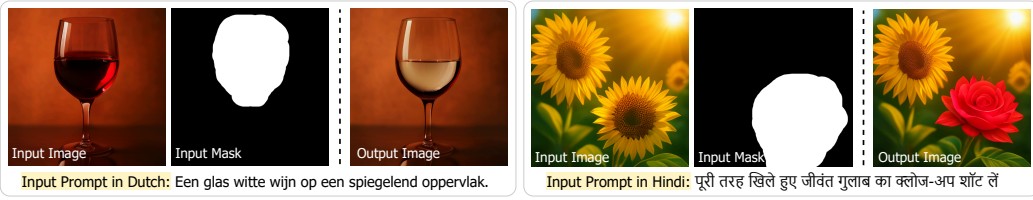

Figure 5: **Multilingual image inpainting**. NEOBABEL supports multilingual text-guided image inpainting, highlighting its potential for interactive visual editing across diverse user groups.

## 6.3 ABLATIONS AND ANALYSES

**Effect of progressive pretraining.** We analyze the impact of our progressive pretraining strategy across three stages at $256 \times 256$ resolution. Progressive pretraining steadily improves multilingual performance (Figure 7; full results in Figure 11 in the appendix). In stage one, training on m-

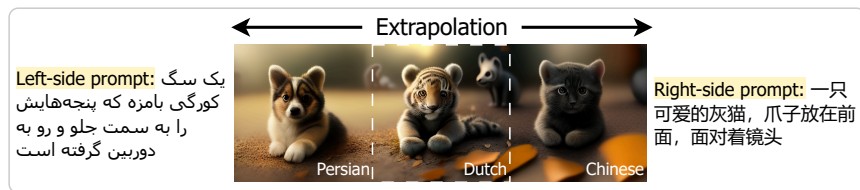

Figure 6: **Multilingual image extrapolation.** NEOBABEL performs text-guided image extrapolation, generating coherent left and right extensions from different multilingual prompts.

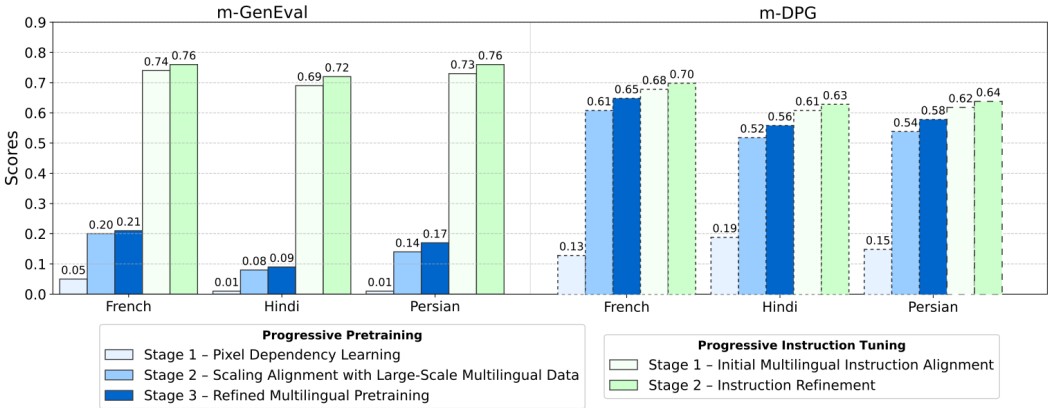

Figure 7: **Effect of progressive pretraining and instruction tuning.** Performance on m-GenEval (left) and m-DPG (right) improves steadily across pretraining and instruction tuning stages. See figure 11 in appendix for results of all six languages on both benchmarks.

ImageNet 1K produces modest scores, reflecting weak multilingual alignment. In stage two, the addition of large but noisy datasets (m-SA-1B, m-CC12M, m-LAION-Aesthetic) drives a significant increase, with gains larger on m-DPG than on m-GenEval, suggesting improved handling of natural multilingual prompts. Stage three incorporates higher-quality datasets (m-LAION-Aesthetic, m-JourneyDB), leading to further improvements. These results highlight the cumulative benefits of progressively increasing both dataset diversity and quality.

**Effect of progressive instruction tuning.** We analyze the effect of high-resolution instruction tuning at $512 \times 512$ resolution using a fixed dataset mixture of m-LAION-Aesthetic, m-JourneyDB, and m-BLIP3o-Instruct. Results are shown in Figure 7, with the full evaluation provided in Figure 11 in the appendix. In the first stage, each training batch consists of 60% m-LAION-Aesthetic, 30% m-JourneyDB, and 10% m-BLIP3o-Instruct samples. This stage yields a substantial multilingual gain on m-GenEval and m-DPG compared to the final stage of pretraining. In the second stage, each training batch shifts emphasis toward higher-quality and instruction-aligned data, consisting of 25% m-LAION-Aesthetic, 60% m-JourneyDB, and 15% m-BLIP3o-Instruct samples. This leads to further multilingual gains in m-GenEval and m-DPG. These results show that beyond increasing the resolution, the relative weight of curated and instruction-focused datasets plays a pivotal role in shaping multilingual capability.

**Cross-lingual image generation.** A more challenging evaluation of the model's multilingual capability involves prompts that combine multiple languages within the same input. This requires the model to integrate information from different languages into a coherent and semantically accurate image. To test this, we design cross-lingual prompts by splitting a base prompt into three parts and translating each into a different language. As seen in Figure 16, these results highlight the model's cross-lingual alignment, despite not being explicitly trained for this task. To further assess robustness, we introduce the Code-Switching Similarity (CSS) score, which measures visual consistency under intra-prompt language variation. We find that NEOBABEL maintains stable performance when language segments are swapped (see Section A.6 in the appendix for details).

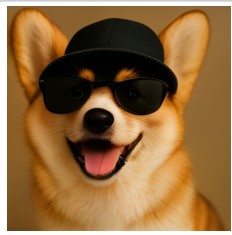

Multilingual Input Prompt: A close-up photograph of a Corgi dog. De hond draagt een zwarte hoed en een ronde, donkere zonnebril. Le Corgi a une expression joyeuse, avec la bouche ouverte et la langue tirée, donnant une impression de bonheur ou d'excitation.

English Translation: A close-up photograph of a Corgi dog. The dog is wearing a black hat and round, dark sunglasses. The Corgi has a joyful expression, with its mouth open and tongue sticking out, giving an impression of happiness or excitement.

Figure 8: **Cross-Lingual Prompt Generation.** An example of a code-switched prompt combining English, Dutch, and French, with the image generated by NEOBABEL. English translations are shown for reader convenience, they are not used as input.

## 7 RELATED WORKS

**Visual generative models.** Two dominant paradigms have emerged for image and video generation: diffusion-based (Rombach et al., 2022; Ramesh et al., 2022; Peebles & Xie, 2023; Bao et al., 2023; Chen et al., 2024a; Xie et al., 2023; Wu et al., 2023a; Lipman et al., 2022; Xie et al., 2025a; Qin et al., 2025; Zhang et al., 2023; Seawead et al., 2025) and autoregressive (Sun et al., 2024; Kondratyuk et al., 2023; Chen et al., 2020; Pang et al., 2024; Li et al., 2025a) models. Diffusion models typically combine pretrained text encoders with denoising networks to iteratively refine visual outputs, while autoregressive models adopt LLM-based architectures trained via next-token prediction. Recent hybrid approaches (Li et al., 2024b; Liu et al., 2024b; Fan et al., 2025) attempt to unify the strengths of both paradigms for more powerful generation. NEOBABEL follows the diffusion-based paradigm but distinguishes itself by adopting an LLM-style architecture for visual token modeling. This removes the reliance on frozen text encoders and instead builds on top of a strong multilingual decoder-based LLM, enabling tighter integration between language and vision.

**Unified multimodal models.** Unified multimodal models aim to handle both image understanding and generation within a single architecture, typically categorized into native and adapter-based approaches. Native approaches such as Chameleon (Chameleon Team, 2024), Show-o (Xie et al., 2025b), Transfusion (Zhou et al., 2025), and Janus (Wu et al., 2025) adopt either autoregressive, diffusion, or hybrid modeling strategies to jointly process vision and language. Recent work (Wang et al., 2024; Wu et al., 2024; Ma et al., 2025; Jiao et al., 2025; Chen et al., 2025b; Song et al., 2025) has focused on improving tokenization and training efficiency to enhance cross-modal alignment. A parallel direction (Tang et al., 2023; Lu et al., 2023; Dong et al., 2024; Ge et al., 2024; Tong et al., 2024; Pan et al., 2025; Chen et al., 2025a; Wu et al., 2023b) constructs unified multimodal models by connecting pretrained LMMs and generative models via adapters or learnable tokens. While modular and flexible, these systems often rely on frozen components and lack full cross-modal integration. Our model, NEOBABEL, aligns more closely with native unified multimodal models by unifying visual and textual modeling within a single decoder-based architecture, without relying on adapters or frozen backbones. Although NEOBABEL supports multilingual multimodal understanding, this work focuses specifically on multilingual image generation.

## 8 CONCLUSION

NEOBABEL demonstrates that high-quality, efficient multilingual image generation is not only possible but also advantageous. Through strategic data curation and a unified architecture, we set a new Pareto frontier in performance, efficiency, and inclusivity.

**Limitations.** While NEOBABEL demonstrates strong multilingual image generation capabilities, several limitations remain. First, the model currently supports six languages; extending to broader linguistic coverage would require tokenizer adaptation and additional training. Second, although NEOBABEL adopts a unified architecture, it does not yet support vision-language tasks such as visual question answering, due to the absence of task-specific fine-tuning. We leave these directions, including task expansion, larger-scale scaling, and broader language coverage, for future research.

**Future work.** This work contributes to the broader goal of democratizing generative AI. By releasing all model weights, datasets, and evaluation protocols, we aim to encourage the research community to build upon this foundation, ultimately advancing toward generative models that better reflect and serve global linguistic and cultural diversity.

**Ethics statement.** All experiments are conducted on *publicly available* datasets used under their original licenses, without collecting new human data or personal annotations. Although the model is not designed to predict demographics or sensitive traits, it may reflect biases in the training data and could be misapplied to privacy-invasive or unauthorized tasks. We caution against such uses and highlight the importance of following license terms, governance standards, and legal requirements.

**Reproducibility statement.** All models, datasets, and baselines in this work are publicly available. We detail training and validation splits, optimization settings, and evaluation protocols in the main text, with further implementation specifics in the appendix. To support replication, we release the full pipeline covering data curation, pretraining, instruction tuning, and evaluation. Training was performed in distributed setups using either 8×NVIDIA H100 (80GB) or 16×AMD MI250x (128GB) GPUs.

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

# A APPENDIX

## A.1 MORE ON RELATED WORKS

**Large multimodal models.** Recent advances in large multimodal models (LMMs) (Liu et al., 2024a; Chen et al., 2024b; Li et al., 2024a; Bai et al., 2025; Dash et al., 2025) have extended large language models (LLMs) (Touvron et al., 2023; Yang et al., 2024) to support image understanding tasks, including image captioning and visual question answering. These models typically rely on a vision encoder to extract image features, which are then projected into the LLM embedding space for cross-modal alignment. More recent encoder-free models (Xie et al., 2025b; Diao et al., 2024; 2025) bypass the explicit image encoder and instead align raw visual tokens directly within the LLM space. Among early efforts to enable multilingual visual understanding, Maya (Alam et al., 2024; 2025), Aya-Vision (Dash et al., 2025) and Pangea (Yue et al., 2024) incorporate a multilingual training corpus. However, they are limited to image understanding tasks. In contrast, our proposed NEOBABEL architecture focuses exclusively on multilingual image generation, offering the first encoder-free model that aligns visual features in the LLM space while supporting cross-lingual generation. Architecturally, NEOBABEL is closely related to show-o (Xie et al., 2025b), sharing the same design goal of direct visual alignment in language space, but differs in its task focus and multilingual design.

## A.2 TRAINING DETAILS

Each stage is trained for 500k steps (except the final stage of instruction tuning with 200k) using the AdamW optimizer and cosine learning rate decay. The learning rate is set to $1e-4$ during pretraining and adjusted during instruction tuning. We gradually increase prompt sequence length and resolution from 128 to 512 and from $256 \times 256$ to $512 \times 512$ respectively. The vocabulary and codebook sizes are fixed across all stages. To further stabilize training and improve generalization, we merge checkpoints across the trajectory, with Simple Moving Average emerging as the most effective strategy (see Sections A.3 and A.4).

Table 4: **Hyperparameters across training progression.**

| Hyperparameters | Pretraining | | | Instruction Tuning | |
|---|---|---|---|---|---|
| | **1st Stage** | **2nd Stage** | **3rd Stage** | **1st Stage** | **2nd Stage** |
| Training Steps | $500k$ | $500k$ | $500k$ | $500k$ | $200k$ |
| Warmup Steps | 5000 | 5000 | 5000 | 5000 | 2000 |
| Learning Rate | $1e-4$ | $1e-4$ | $1e-4$ | $2e-4$ | $5e-05$ |
| Learning Rate Decay | cosine | cosine | cosine | cosine | cosine |
| Optimizer | AdamW | AdamW | AdamW | AdamW | AdamW |
| Image Resolution | $256 \times 256$ | $256 \times 256$ | $256 \times 256$ | $512 \times 512$ | $512 \times 512$ |
| LLM Sequence Length | 128 | 512 | 512 | 512 | 512 |
| LLM Vocab Size | $256k$ | $256k$ | $256k$ | $256k$ | $256k$ |
| Codebook Size | 8192 | 8192 | 8192 | 8192 | 8192 |

## A.3 MULTILINGUAL MODEL MERGING

To enhance generalization and stability of multilingual image generation models, we adopt model merging techniques that combine multiple checkpoints from the training trajectory. Let $\{M_i\}_{i=1}^N$ denote a sequence of $N$ model checkpoints and $\{w_i\}_{i=1}^N$ their corresponding non-negative weights. The merged model $\widehat{M}$ is defined as a convex combination:

$$\widehat{M} = \sum_{i=1}^N \alpha_i M_i \quad \text{where} \quad \alpha_i = \frac{w_i}{\sum_{j=1}^N w_j}. \tag{1}$$

This formulation allows the merged model to interpolate within the solution space spanned by the selected checkpoints, potentially improving generalization on unseen prompts and enhancing robustness to overfitting. We consider three widely used weighting strategies for this purpose, each

reflecting different assumptions about model evolution during training. The comparative results and analysis of these approaches are presented ablation studies section.

**Simple Moving Average** (SMA) assigns equal weight to all checkpoints. It is defined as:

$$M_{\text{avg}} = \frac{1}{N} \sum_{i=1}^{N} M_i. \tag{2}$$

SMA is simple, stable, and particularly effective when applied in the later stages of training where model weights exhibit minimal drift. Prior work (Li et al., 2025c) found SMA to perform robustly due to this stabilization.

**Exponential Moving Average** (EMA) emphasizes recent checkpoints by applying exponentially decaying weights. It is computed recursively as:

$$M_{\text{avg}}^{(i)} = \alpha M_i + (1 - \alpha) M_{\text{avg}}^{(i-1)}, \quad i \in [2, N]. \tag{3}$$

The decay factor $\alpha \in (0, 1)$ controls the trade-off between recency and stability. EMA adapts more quickly to recent model dynamics but is sensitive to noise if weights are unstable.

**Weighted Moving Average** (WMA) assigns custom, possibly increasing weights to later checkpoints. The merged model is computed using the normalized form:

$$M_{\text{avg}} = \sum_{i=1}^{N} \frac{w_i}{w_{\text{sum}}} M_i, \quad \text{where} \quad w_{\text{sum}} = \sum_{i=1}^{N} w_i. \tag{4}$$

This general formulation allows flexibility in how much importance is placed on each checkpoint. In our case, we use $w_i = i$ to emphasize later-stage models.

## A.4 EFFECT OF MODEL MERGING ON GENERALIZATION

We investigate the impact of model merging on multilingual image generation performance by combining $N = 20$ checkpoints sampled at 10,000-step intervals from the second instruction tuning stage (steps 0–200K). Table 5 reports the results of three merging strategies: Simple Moving Average (SMA), Exponential Moving Average (EMA), and Weighted Moving Average (WMA) compared to the last checkpoint baseline. As reported, m-GenEval score on English prompt improves from 0.81 to 0.83 after model merging. Both WMA and SMA reach this upper bound, indicating that merging checkpoints along the optimization path enhances semantic alignment. More-

Table 5: **Effect of model merging on generalization.** All merging strategies improve performance on m-GenEval and slightly enhance m-DPG, highlighting model merging as a simple yet effective way to boost generalization.

| Method | m-GenEval | m-DPG |
|---|---|---|
| Last checkpoint | 0.81 | 0.73 |
| EMA | 0.82 | 0.75 |
| WMA | 0.83 | 0.75 |
| SMA | **0.83** | **0.75** |

over, m-DPG score on English prompt remains stable or show modest gains, suggesting that merging preserves the model's ability to accurately follow dense, attribute-rich prompts without sacrificing fine-grained multilingual grounding. Among the merging strategies, SMA performs best overall due to its uniform averaging over well-aligned checkpoints. EMA also improves results but remains more susceptible to short-term training noise. WMA offers a compromise by emphasizing later checkpoints, trading off stability for adaptability. These findings underscore that checkpoint merging can meaningfully enhance both compositional understanding and multilingual robustness, with SMA offering a simple yet effective strategy.

## A.5 DATA FILTERING

In this section, we describe in detail the four post-processing and filtering strategies: length filtering, language validation, visual–text mismatch filtering, and toxicity/NSFW filtering, used to ensure dataset quality and consistency across languages.

- **Length filtering**: Remove captions that are too short (e.g., fewer than 5 tokens) or excessively long (e.g., more than 500 tokens).

- **Language validation**: Detect and discard captions containing non-English phrases or corrupted outputs using language identification tools. We use the `fastText` language identification model trained on 176 languages (Joulin et al., 2016). We discard any caption not classified as English with a confidence score above 90%.

- **Visual-text mismatch filtering**: Discard captions that do not align with visual content, measured via auxiliary vision-language models (e.g., using VQAScore). Specifically, we leverage MolMo-72B (Deitke et al., 2025) deployed with vLLM (Kwon et al., 2023), formulating the task as a binary structured prediction (`yes/no`) via vLLM's output interface.

- **Toxicity and NSFW filtering**: Discard samples using the LAION-5B NSFW classifier (Schuhmann et al., 2022) to ensure safe visual content before captioning, assuming high likelihood of appropriateness in the resulting captions.

### A.6 CODE-SWITCHING SIMILARITY (CSS)

**Definition.** A multilingual model should demonstrate robustness not only to monolingual prompts but also to mixed-language inputs i.e. to code-switching. Code switching often increases perplexity and degrades performance in language models; however, its impact on image generation remains largely unexplored. To evaluate this, we introduce the CSS Score, which quantifies visual consistency under intra-prompt language variation. Given a set of reference prompts in English, we construct two variants per prompt for each of the $L-1$ non-English target languages: (1) `English-First (EF)`: the first half of the prompt remains in English while the second half is translated into the target language, and (2) `English-Second (ES)`: the first half is translated while the second half remains in English. For each prompt $p \in \{p_i\}_{i=1}^{P}$, we generate a single reference image $x_{\text{ref}}$ from the original English prompt and $L-1$ code-switched images: $x_{\text{EF}}^{(l)}$ and $x_{\text{ES}}^{(l)}$ for each target language $l$. Each image is encoded into an embedding $f(x) \in \mathbb{R}^d$ using a vision encoder. The Code Switching Similarity (CSS) score for each prompt is computed by measuring the average cosine similarity between the reference embedding $f(x_{\text{ref}})$ and the embeddings from the EF and ES variants:

$$\text{CSS}p^{\text{EF}} = \frac{1}{L-1} \sum_{l=1}^{L-1} \cos\left(f(x_{\text{ref}}), f(x_{\text{EF}}^{(l)})\right), \quad \text{CSS}p^{\text{ES}} = \frac{1}{L-1} \sum_{l=1}^{L-1} \cos\left(f(x_{\text{ref}}), f(x_{\text{ES}}^{(l)})\right).$$

(5)

The final CSS scores are obtained by averaging across all prompts. To assess how well models preserve semantic consistency under intra-prompt code switching, we report both $\text{CSS}^{\text{EF}}$ and $\text{CSS}^{\text{ES}}$, using embeddings from EVA-CLIP (Sun et al., 2023b) and DINOv2 (Oquab et al., 2023) computed on m-DPG prompts.

**Analysis.** We assess the CSS score with EVA-CLIP and DINOv2. As shown in Table 6, NEOBABEL outperforms larger models under both English-First (EF) and English-Second (ES) prompts. EVA-CLIP scores show minimal EF–ES differences, indicating limited impact of English segment position on semantic alignment, while DINOv2 scores are consistently lower, reflecting the greater challenge of maintaining structural coherence under language mixing. A robust model should achieve both high CSS and small EF–ES gaps, a balance met by NEOBABEL with 0.82/0.81 on EVA-CLIP and 0.67/0.64 on DINOv2. Figure 9 further shows that, unlike BLIP3-o (8B) which exhibits high variance across prompts, NEOBABEL combines higher medians with lower dispersion, confirming its consistent handling of code-mixed inputs. These results demonstrate that effective multilingual alignment, not parameter count, is key to robustness under code switching.

### A.7 USE OF LARGE LANGUAGE MODELS

We used large language models exclusively for grammar correction and language refinement of the manuscript. It played no role in research ideation, methodological design, experimentation, data analysis, or result generation; all technical content was solely developed and validated by the authors.

Table 6: **Code Switching Similarity (CSS) analysis** using EVA-CLIP and DINOv2 backbones. Scores are reported for two prompt variants: English First (EF) and English Second (ES). NEO-BABEL (2B) outperforms larger models, showing strong visual consistency and robustness to code-mixed input order. The larger DINOv2 gap reflects its higher sensitivity to visual-structural variation, while EVA-CLIP remains more stable due to its semantic focus.

| Model | Params | EVA-CLIP | | DINOv2 | |
|---|---|---|---|---|---|
| | | EF | ES | EF | ES |
| Show-o | 1.3B | 0.73 | 0.72 | 0.41 | 0.38 |
| Janus | 1.3B | 0.75 | 0.73 | 0.50 | 0.43 |
| Janus Pro | 7B | 0.76 | 0.72 | 0.58 | 0.50 |
| BLIP3-o | 4B | 0.75 | 0.75 | 0.54 | 0.54 |
| BLIP3-o | 8B | 0.74 | 0.74 | 0.52 | 0.51 |
| NEOBABEL | 2B | **0.82** | **0.81** | **0.67** | **0.64** |

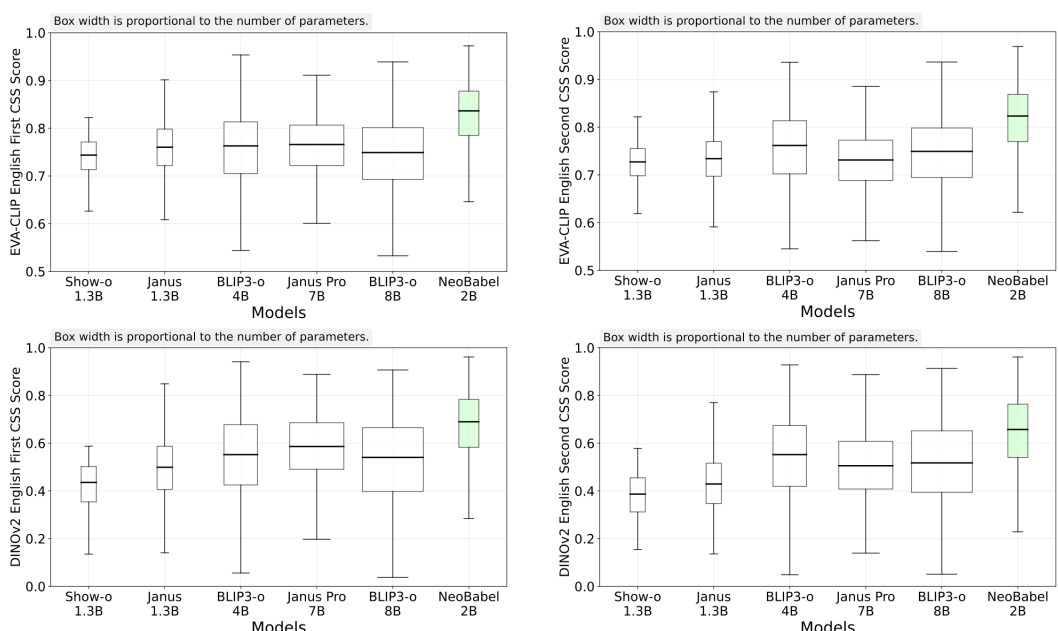

Figure 9: **Variation in Code Switching Similarity (CSS) scores across models.** We report CSS scores for code-mixed prompts under two settings: English-first (left column) and English-second (right column), using EVA-CLIP (top row) and DINOv2 (bottom row) as backbones. Higher scores indicate stronger visual alignment with the reference image, while smaller EF–ES gaps suggest robustness to code-switch position. NEOBABEL consistently achieves higher medians and lower variance than larger baselines, especially under DINOv2, highlighting its effective and stable handling of multilingual prompts.

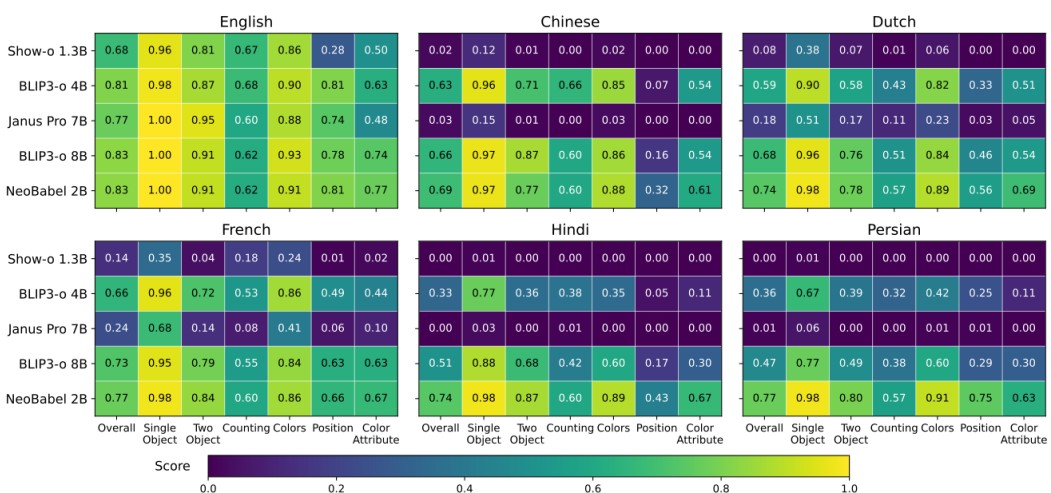

Figure 10: **m-GenEval benchmark comparison.** Models such as Janus Pro and BLIP3-o rely on multilingual base LLMs but are trained solely on English image-generation data, leading to a sharp performance drop in non-English languages. In contrast, NEOBABEL maintains strong and consistent results across all languages, demonstrating robust cross-lingual generalization.

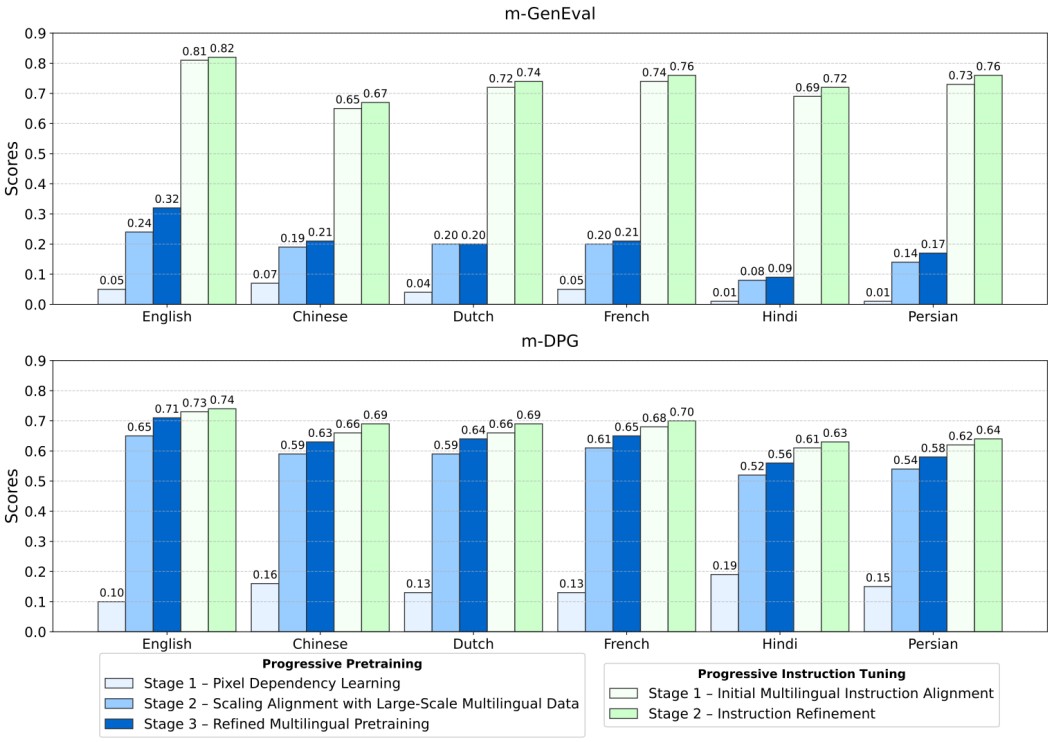

Figure 11: **Effect of progressive pretraining and instruction tuning.** Performance on m-GenEval (top) and m-DPG (bottom) improves steadily across pretraining and instruction tuning stages.

Hindi: चित्र एक शैलीबद्ध, कार्टून शैली के कंकाल चरित्र को चित्रित करता है, जो एक जादूगर या जादूगर के रूप में पहना हुआ है, जो दृश्य के कुछ हिस्सों को उजागर करने वाले नीले प्रकाश के साथ नीले प्रकाश के उच्चारण के साथ एक अंधेरे पृष्ठभूमि के खिलाफ खड़ा है। यह आंकड़ा एक ओवरसाइज़ हुड ड्रेस पहनता है जो अपने सिर और कंधों को पूरी तरह से कवर करता है, केवल खोपड़ी को दोनों तरफ दो बड़े आंखों के सॉकेट के माध्यम से दिखाई देता है, जहां आंखें स्थित होंगी। चेहरा कंकाल जैसा दिखता है लेकिन आंखों के लिए इन छेदों के अलावा कोई अन्य चेहरे की विशेषताएं नहीं हैं। इस पोशाक में मध्ययुगीन शैली के कपड़े की तरह दिखने के लिए व्यवस्थित हड्डी के टुकड़ों से बने कवच जैसे टुकड़े शामिल हैं; यह विभिन्न हड्डों से सजाया गया है, जिसमें कंधे के पैड जैसे कंधे के पैड शामिल हैं, जो उनके केंद्र में स्टार मोटिवों से सजाए गए ढाल के रूप में ढाल के रूप में सजावट के तत्व हैं। बटन जैसे सजावटी तत्व भी हैं।

این تصویر یک اثر هنری بسیار دقیق و سورئال است که نمای نیمرخ صورت زنی را به تصویر می‌کشد و عناصر پیچیده‌ی ارگانیک در ویژگی‌های چهره او ادغام: Persian شده‌اند. پوست او با بافت‌های متنوعی شبیه برگ‌ها، تاک‌ها و الگوهای ظریف تزئین شده که به‌طور یکپارچه با طرح‌های طبیعی مانند گل‌ها و پروانه‌ها ترکیب می‌شوند. چشم او حالت اثیری دارد، با عنبیه‌های آبی درخشان که با درخشش طلایی و مژه‌های بلند احاطه شده‌اند. لب‌هایش به‌دلیل اغراق هنری برای ایجاد اثر دراماتیک، پرتر از حالت عادی هستند. ترکیب‌بندی شامل رنگ‌های زنده‌ای چون طلایی، آبی، سبز، بنفش و رنگ‌های خاکی است که با لایه‌های مختلف نقاشی عمق ایجاد می‌کند

Dutch: De afbeelding toont een prachtig vormgegeven glazen fles gevuld met een levendige groene vloeistof, die magisch en betoverd oogt. De fles is aan de onderkant ingewikkeld vormgegeven met sierlijke patronen die een antieke stijl suggereren. Vanuit het binnenste ontspruiten weelderige wijnranken, versierd met bladeren in verschillende vormen en maten, waarvan sommige lijken op klimopachtige bladeren die om de hals verstrengeld zijn. Binnenin de fles wervelt de inhoud met gloeiende, smaragdgroene energie, wat een dynamische beweging creëert alsof de fles leeft. Kleine belletjes zweven omhoog door de vloeistof en voegen diepte en dynamiek toe aan de scène. Een paar kleine voorwerpen zoals munten en stukjes papier liggen verspreid op het oppervlak onder de fles, wat een gevoel van mysterie suggereert over hun oorsprong of doel. De belichting werpt zachte schaduwen over deze elementen, wat de mystieke sfeer versterkt en tegelijkertijd de ingewikkelde details zowel binnen als buiten de fles benadrukt. Over het geheel genomen is er een fantasie-element dat doet denken aan alchemie of het maken van toverdranken, wat nieuwsgierigheid en verwondering oproept.

French: La photo montre une jeune femme aux cheveux foncés, coiffée avec élégance, ornée d'une tiare complexe qui scintille subtilement sur son front et ses tempes. Son maquillage est discret mais raffiné, mettant en valeur ses grands yeux grâce à des sourcils bien dessinés. Elle porte une robe luxueuse en tissu riche, avec des broderies élaborées et des perles sur le corsage, formant des motifs gracieux au niveau de la poitrine. Les manches ajustées de la robe sont décorées de détails floraux délicats sur les épaules, ajoutant à son élégance. Un élément remarquable comprend des décorations semblables à des plumes s'étendant de son encolure jusqu'à sa clavicule, lui donnant une allure royale, comme si elles faisaient partie intégrante du vêtement plutôt que d'être des accessoires. Son expression est dramatique et gracieuse, sur un fond sombre et serein qui met en valeur la douceur des textures.

Figure 12: **Qualitative evaluation of NEOBABEL.** Each row is based on a single concept expressed in six different languages. We show only one of the prompts (in one language) and present six images generated from its translated prompts in the other five languages. Across all languages, NEOBABEL delivers semantically accurate and visually cohesive outputs with reliable consistency.

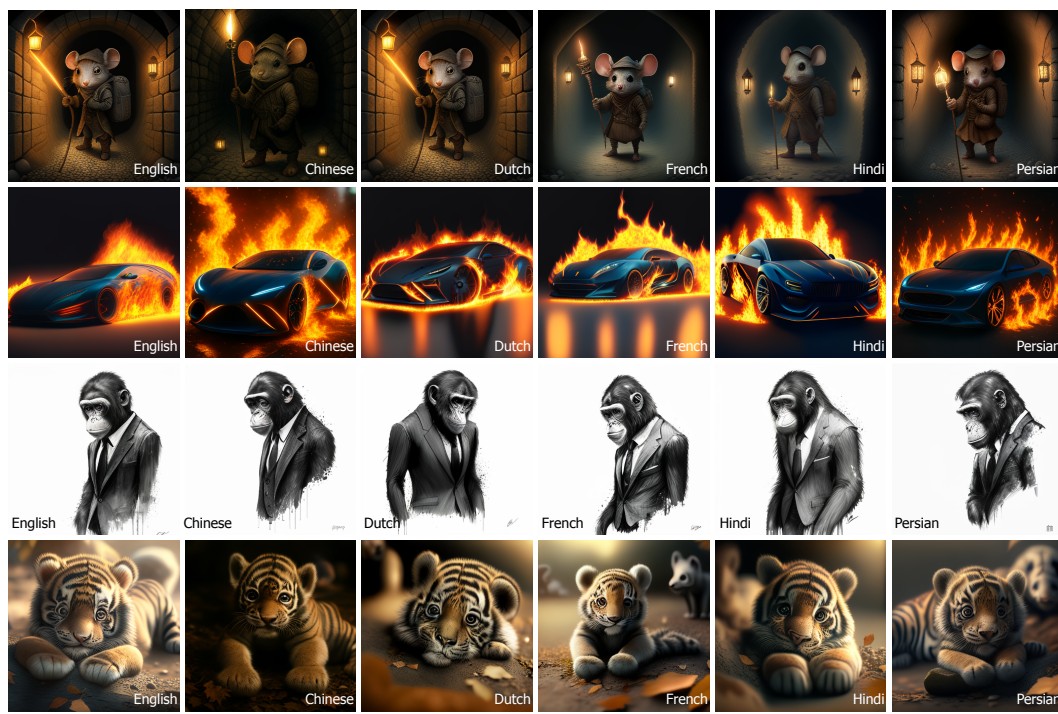

Figure 13: **Qualitative evaluation of NEOBABEL.** Each row corresponds to a single concept expressed in six different languages: English, Chinese, Dutch, French, Hindi, and Persian. Although prompts are not shown for readability, all images were generated using translated versions of the same underlying prompt in each language. NEOBABEL consistently produces semantically aligned and visually coherent results across languages, highlighting its strong multilingual generation capabilities. We intentionally omit the prompts here due to their length, focusing instead on the visual consistency across languages.

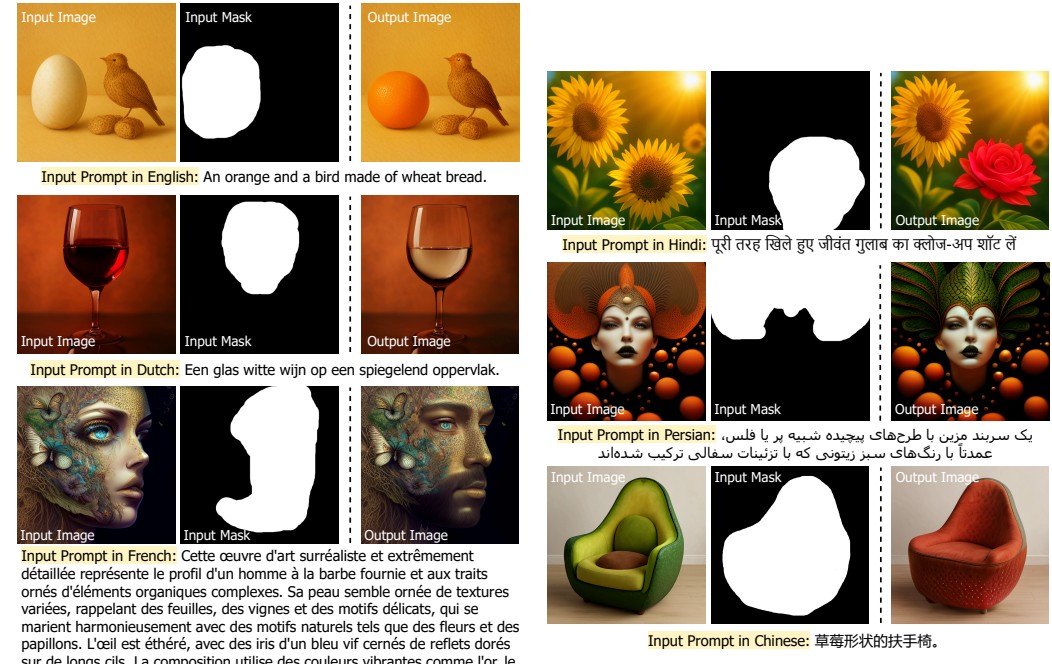

Figure 14: **Multilingual image inpainting**. NEOBABEL supports multilingual text-guided image inpainting, highlighting its potential for interactive visual editing across diverse user groups.

Figure 15: **Multilingual image extrapolation.** NEOBABEL performs text-guided image extrapolation, generating coherent left and right extensions from different multilingual prompts.

Figure 16: **Cross-Lingual Prompt Generation.** An example of a code-switched prompt combining English, Dutch, and French, with the image generated by NEOBABEL. English translations are shown for reader convenience, they are not used as input.