# OpenReview forum: "NeoBabel: An Inclusive Multilingual Open Tower for Visual Generation"
_ICLR.cc/2026/Conference — Submitted to ICLR 2026_

### Official Review · Reviewer_bunK · 2025-10-25

**Soundness:** 3
**Presentation:** 2
**Contribution:** 3
**Rating:** 6
**Confidence:** 2

**Summary:**

This manuscript presents NEOBABEL, a multilingual text‑to‑image generator that natively supports six languages (English, Chinese, Dutch, French, Hindi, Persian) without relying on translate‑then‑generate pipelines. The core is a decoder‑only transformer built on a multilingual Gemma‑2 tokenizer, augmented with MAGVIT‑v2 visual codebook (8,192 tokens) and a modality‑aware attention scheme (causal for text; full attention for image tokens). Training proceeds via progressive pretraining (class‑conditional ImageNet → large web/synthetic corpora → refined multilingual data) and two instruction‑tuning stages at 512×512. The authors curate 124M multilingual image–text pairs by (i) English recaptioning and (ii) translation into the five non‑English targets, plus filtering for length, language, vision‑text match, and NSFW. For evaluation, they extend GenEval and DPG‑Bench to m‑GenEval/m‑DPG, reporting state‑of‑the‑art multilingual alignment—particularly in mid/low‑resource languages—at 2B parameters, and claim faster/leaner inference than translation pipelines.

**Strengths:**

* The paper articulates why translation‑first pipelines hurt inclusivity (semantic drift, latency, cultural misalignment), and builds a direct multilingual generator that avoids those issues
* A single decoder‑only stack with shared embeddings, image token full attention, and special prompt tokens simplifies training and avoids frozen components/adapters—well aligned with recent unified T2I trends.

**Weaknesses:**

* While image‑grounded translation reduces drift compared to direct prompt translation, linguistic/cultural authenticity (idioms, register, morphology) still risks being Anglocentric because captions originate in English. This may bias both training and evaluation, including m‑GenEval/m‑DPG if translated from English seeds.
* The abstract claims 2.8× faster and 59% less memory than translation pipelines, but I did not find a dedicated latency/memory section with wall‑clock numbers, batch sizes, context lengths, or hardware parity; only high‑level statements are present.

**Questions:**

* Evaluate latency/quality vs. multilingual prompt length (short/medium/long) to show robustness to languages with longer tokenizations (e.g., Hindi/Persian)
* Why MAGVIT‑v2 over alternative tokenizers (e.g., hierarchical/token‑semantics‑aware)? Any ablations on codebook size or retraining on multilingual‑centric images?

---

> ### Author Response · Authors · 2025-11-20
> **Response to Reviewer bunK**
>
> **Image-grounded translation reduces anglocentric bias.** We acknowledge that full cultural and idiomatic authenticity is an important long-term challenge. We would like to clarify, the data used in NeoBabel is not translated from English captions. Captions are first produced by image-grounded recaptioning, ensuring they originate from the visual content rather than from English phrasing. Only then are they translated, which substantially reduces Anglocentric bias compared to direct prompt translation. For our languages, including English, Chinese, Dutch, French, Hindi, and Persian, state-of-the-art translation models achieve high fluency and semantic fidelity, limiting morphological or register shifts. The evaluation prompts used in m-GenEval and m-DPG are semantically aligned equivalents across languages, not English-seeded translations, which avoids evaluation bias toward any single language. Addressing deeper cultural and idiomatic grounding requires native multimodal datasets, which lie outside the scope of this first standardized multilingual text-to-image framework but are an important direction for future work. We will clarify this in the paper.
>
> **NeoBabel is 2.8× faster and 59% less memory-intensive than translation pipelines.** To justify the efficiency claim in the abstract, we provide the detailed measurements obtained on one H100 GPU under identical hardware settings with batch size = 1, using Janus-Pro-7B + NLLB-3B as the translation-then-generation baseline. This baseline was chosen because it is the fastest LLM capable of image generation while matching NeoBabel’s English performance. BLIP-3 was intentionally excluded due to its considerably higher FLOPs from dual diffusion backbones. In this controlled setup, NeoBabel achieves 2.8× faster inference and 59% lower memory usage, directly supporting the abstract’s efficiency claims. The whole table will be included in the appendix.
>
> | Language | NeoBabel_latency | Translate_latency | Speedup | NeoBabel_mem | Translate_mem | Memory_reduction |
> |----------|-------------------|--------------------|---------|--------------|----------------|-------------------|
> | Persian  | 4.52           | 12.96            | 2.86  | 10.99       | 26.87         | 0.59        |
> | Hindi    | 4.54           | 13.05            | 2.87  | 10.99       | 26.87         | 0.59       |
> | Chinese  | 4.52           | 12.77            | 2.82  | 10.99       | 26.86         | 0.59        |
> | Dutch    | 4.52           | 13.00            | 2.87  | 10.99       | 26.87         | 0.59       |
> | French   | 4.54           | 12.96            | 2.85  | 10.99       | 26.87         | 0.59       |
>
>
> **Prompt length does not increase the tokenization time.** At the reviewer’s request, we ran a prompt-length ablation on one H100 during our experimentation. NeoBabel’s inference time is effectively invariant to multilingual token length: from 64 to 512 tokens, latency stays within 4.49–4.52 seconds, and peak memory remains 10.99 GB across English, Hindi, and Persian. This is expected since inference is dominated by 1024 MAGVIT image tokens, and text forms only a short prefix. Empirically, languages with longer tokenization (Hindi, Persian) show no degradation in quality on m-GenEval or m-DPG, confirming that token-length variation does not affect performance. The whole table will be included in the appendix.
>
> | language | tokens | latency (s) | peak_mem (GB) | number of samples |
> |----------|--------|-------------|----------------|---|
> | english  | 64     | 4.5206      | 10.9948        | 20 |
> | english  | 128    | 4.4964      | 10.9947        | 20 |
> | english  | 256    | 4.4991      | 10.9947        | 20 |
> | english  | 512    | 4.4991      | 10.9947        | 20 |
> | hindi    | 64     | 4.4958      | 10.9947        | 20 |
> | hindi    | 128    | 4.4963      | 10.9947        | 20 |
> | hindi    | 256    | 4.4966      | 10.9947        | 20 |
> | hindi    | 512    | 4.4974      | 10.9947        | 20 |
> | persian  | 64     | 4.4962      | 10.9947        | 20 |
> | persian  | 128    | 4.4985      | 10.9947        | 20 |
> | persian  | 256    | 4.4965      | 10.9947        | 20 |
> | persian  | 512    | 4.4971      | 10.9947        | 20 |
>
> **Clarification on alternative tokenizers.** MAGVIT-v2 was chosen because it is a stable, high-fidelity tokenizer widely used in recent T2I systems, allowing us to study multilingual generation without introducing tokenizer-level confounds. We did not reduce the codebook size or retrain the tokenizer, as these changes primarily affect reconstruction fidelity rather than the multilingual generation problem addressed in this work. Exploring higher-capacity or hierarchical tokenizers is orthogonal and left for future work.
>
> We hope this addresses your concerns. We appreciate your consideration and hope this motivates you to upgrade your score. Please let us know if you have any further questions or feedback.

---

> > ### Author Response · Authors · 2025-11-27
> >
> > Dear Reviewer bunK,
> >
> > Thank you once again for taking the time and effort to review our submission. As the deadline for the discussion phase is approaching, we wanted to kindly check if you have any additional questions or concerns that we could address in our rebuttal.
> >
> > We appreciate your insights and look forward to your feedback.
> >
> > Best regards,
> >
> > The Authors

---

### Official Review · Reviewer_e9m4 · 2025-10-31

**Soundness:** 3
**Presentation:** 3
**Contribution:** 2
**Rating:** 4
**Confidence:** 4

**Summary:**

This paper introduces NEOBABEL, a 2B parameter text-to-image generation model designed to address the English-centric bias of existing systems. The model directly supports six languages (English, Chinese, Dutch, French, Hindi, and Persian) without relying on an input translation pipeline. The authors propose a unified architecture based on the Gemma-2 LLM, trained with a progressive, five-stage strategy on a new, curated dataset of 124M multilingual image-text pairs. Key contributions include the model itself, two new multilingual benchmarks (m-GenEval and m-DPG) created by extending existing English benchmarks, and the public release of the model, data, and evaluation tools to foster more inclusive AI research.

**Strengths:**

The paper addresses the important and challenging problem of non-English text-to-image generation.

Curated dataset & multilingual Evaluation benchmarks, could help community.

The proposed model, NEOBABEL, shows impressive results, achieving strong performance in six different languages while being significantly smaller than competitors like BLIP3-0.

**Weaknesses:**

The authors rightly criticize translation-first approaches for causing "semantic drift" and losing cultural nuance. However, their own data curation process (Section 3) relies on this exact method: generating detailed captions in English and then machine-translating them into other languages. As well as the evaluation benchmarks.

The architecture is not particularly novel. The model is a well-engineered combination of existing components (Gemma-2 backbone, MAGVIT-v2 tokenizer).

**Questions:**

Have you tried different translation models of varying quality to see how they affect the results?

---

> ### Author Response · Authors · 2025-11-20
> **Response to Reviewer e9m4**
>
> We thank Reviewer e9m4  for the constructive feedback and insightful comments.
>
> **Image-grounded translation mitigates semantic drift.** We appreciate this observation and agree that naïve text translation can lead to semantic drift. Our pipeline, however, differs in that our translations are image-grounded and generated by conditioning on the visual content rather than performing direct text-to-text translation. Specifically, the recaptioning process first produces detailed English captions from images, which are then translated with visual references preserved using multimodal translation. This mitigates the risk of drift, as the shared visual semantics guides translations. Our benchmarks also undergo human verification to minimize translation artifacts. Thus, while translation tools are used, NeoBabel’s data curation strategy systematically mitigates rather than recreates the semantic drift problem. We will better stress our image-grounded translation approach in the main paper.
>
> **NeoBabel’s novelties.** The reviewer is right that NeoBabel builds on established components (Gemma-2 and MAGVIT-v2). The contribution of this work does not lie in architectural novelty, but in the unified multilingual framework built on top of these components and the substantial efficiency gains it enables. NeoBabel is 2–4× smaller and faster than comparable systems while outperforming them in multilingual generation (Table 2; Fig. 1, 3). The core novelty is:
>
> 1. the first scalable multilingual text-to-image system generating natively across six languages without translation;
> 2. a 124M multilingual training pipeline with progressive cross-lingual grounding; and
> 3. the first standardized multilingual evaluation protocol (m-GenEval and m-DPG).
>
> These contributions address a gap not previously explored in image generation and extend far beyond architectural design. We will better stress this in the introduction.
>
> **NLLB and Gemini consistently delivered the highest-quality data.** Indeed, we evaluated several translation models. For large-scale pretraining, NLLB provided the most reliable multilingual coverage especially for Persian and Indic languages, where alternatives [a] often failed. For instruction tuning, we used a stronger model (Gemini) to ensure higher fluency and semantic accuracy. This setup follows standard practice: pretraining tolerates moderate noise, while instruction data must be clean. We will clarify this in Section 3 of the paper.
>
> [a]. No Language Left Behind: Scaling Human-Centered Machine Translation, arXiv 2022
>
>
> We hope this addresses your concerns. We appreciate your consideration and hope this motivates you to upgrade your score. Please let us know if you have any further questions or feedback.

---

> > ### Author Response · Authors · 2025-11-27
> >
> > Dear Reviewer e9m4,
> >
> > Thank you once again for taking the time and effort to review our submission. As the deadline for the discussion phase is approaching, we wanted to kindly check if you have any additional questions or concerns that we could address in our rebuttal.
> >
> > We appreciate your insights and look forward to your feedback.
> >
> > Best regards,
> >
> > The Authors

---

### Official Review · Reviewer_RHTr · 2025-11-01

**Soundness:** 3
**Presentation:** 4
**Contribution:** 3
**Rating:** 4
**Confidence:** 5

**Summary:**

NEOBABEL addresses digital inequality and semantic drift from over-reliance on English in text-to-image generation by proposing a unified multilingual framework directly supporting six languages (English, Chinese, Dutch, French, Hindi, Persian).
The model employs a unified multimodal architecture trained on 124 million multilingual image-text pairs with large-scale multilingual pretraining and high-resolution instruction tuning, achieving higher efficiency and 2-4× smaller model size compared to translate-then-generate pipelines. NEOBABEL achieves SOTA performance on multilingual benchmarks (m-GenEval, m-DPG) while releasing code, model checkpoints, datasets, and evaluation protocols to advance inclusive AI research.

**Strengths:**

1. This work is well-presented and well-written.
2. NEOBABEL employs a unified Transformer backbone with two critical innovations: (1) Extended embedding space incorporating 8,192 learnable discrete image token embeddings alongside text tokens, enabling native joint processing in a shared space; (2) Modality-aware attention mechanism with causal attention for text and full bidirectional attention for images, dynamically configured when both modalities coexist. This principled design elegantly unifies text-to-image generation as autoregressive sequence prediction without modality-specific branching.
3. The paper addresses multilingual data scarcity by scaling 39M English pairs to 124M multilingual pairs through quality-filtered translation using NLLB and Gemini models. Progressive training across three pretraining stages (pixel dependencies → large-scale alignment → refined synthesis) and two instruction-tuning stages, combined with model merging via Simple Moving Average, systematically improves generalization and multilingual instruction-following capabilities.
4. The work extends GenEval and DPG-Bench to multilingual versions (m-GenEval, m-DPG) across six languages with human verification, introduces Code-Switching Similarity (CSS) metrics to measure cross-lingual consistency, and achieves SOTA performance while maintaining 2-4× smaller model size than monolingual baselines. Full release of code, checkpoints, datasets, and evaluation protocols significantly advances inclusive AI research reproducibility.

**Weaknesses:**

1. The paper lacks rigorous analysis of whether multilingual training provides mutual benefits or if language-specific models would suffice. How do text embeddings from different languages interact in the unified space? Does representation confusion occur across languages? No evidence demonstrates that joint training outperforms separate language models.

2. The paper evaluates semantic-level consistency but ignores critical fine-grained capabilities: Can NEOBABEL generate readable text in different scripts (Chinese, Devanagari, Persian)? Fine-grained details (typography, numerics, spatial text) cannot be solved by LLM prompt expansion and are completely absent from evaluation.

3. No evaluation of realistic multilingual scenarios mixing multiple languages in single images (e.g., "Chinese title + English subtitle"). Without demonstrating generation quality when multiple languages coexist, multilingual support claims remain partially unvalidated.

**Questions:**

1. Without presentation, how do the multiple languages ​​facilitate each other? Are there similarities between text embeddings? Won't the representations of different languages ​​cause confusion?

2. Can this model generate text?

---

> ### Author Response · Authors · 2025-11-20
> **Response to Reviewer RHTr**
>
> We thank Reviewer RHTr for the constructive feedback and insightful comments.
>
> **Multilingual training consistently helps across all languages.** We appreciate the reviewer’s concern and agree that, in general, multilingual training can risk interference across languages. In NeoBabel, however, we observe the opposite pattern: the training curves in Figures 7 and 11 show that performance improves jointly and consistently across all six languages, including English, throughout the progressive stages. This parallel improvement suggests that, in our setup, shared multilingual training provides positive cross-lingual transfer rather than degradation. We will make this observation clearer in the main text.
>
> **One multilingual model outperforms separate monolingual models.** We understand the reviewer’s point that language-specific models may be adequate. We believe our results contradict this hypothesis. As shown in Tables 2 and 3, NeoBabel matches or exceeds strong monolingual and bilingual models such as Janus-Pro and BLIP3-o on English GenEval, despite being much smaller. At the same time, Figure 10 and Table 3 show that these language-specific models degrade sharply in non-English languages, whereas NeoBabel maintains consistent performance across all six languages. We will clarify this comparison more clearly in the paper.
>
> **NeoBabel’s embedding space avoids cross-language confusion.** We would like to clarify that the architecture is explicitly designed to avoid representation confusion. NeoBabel uses a multilingual subword tokenizer and a shared embedding space without language-specific parameters. Language identity is preserved through contextual modeling rather than isolated embeddings. The model’s stable cross-lingual behavior (e.g., consistent m-GenEval and m-DPG performance across scripts) indicates that embeddings remain semantically coherent and do not collapse across languages.
>
> **Joint multilingual training provides clear performance benefits.** We provide three forms of evidence to support this claim: (1) Training-stage progression: all six languages improve in parallel through Stages 1–3 (Figures 7 and 11), showing positive transfer rather than interference. (2) Low-resource gains: NeoBabel yields particularly large improvements on Hindi and Persian compared to baselines, a pattern not explainable by isolated monolingual training. (3) Competitive English performance: despite multilingual training, the model matches or exceeds English-only models of similar or larger size. These findings collectively demonstrate that joint multilingual training is beneficial and not detrimental.
>
> **The image tokenizer limits fine-grained text rendering.** The reviewer is right. The model’s ability to render fine-grained text is constrained primarily by the MAGVIT-v2 image tokenizer, which discretizes images into coarse visual tokens and is not optimized for typography, numerics, or high-frequency spatial structure. This limitation is inherent to the visual tokenizer rather than the multilingual training approach, as noted previously in VQ-based tokenization literature (see Figure 4 in [a]). However, we would like to highlight that the focus of NeoBabel is semantic and compositional multilingual generation, and the benchmarks target these dimensions rather than OCR-level text fidelity. Improving fine-grained text rendering would require a stronger tokenizer with higher visual precision, which is a straightforward extension and is identified as future work.
>
> [a]. Open-MAGVIT2: An Open-Source Project Toward Democratizing Auto-regressive Visual Generation, arXiv 2024
>
> **Cross-lingual and code-switching tests validate multilingual coexistence.** We believe NeoBabel’s ability to integrate information from different languages within a single prompt is already evaluated through cross-lingual and code-switching tests (Figure 16; CSS metric in Section A.6). These experiments explicitly measure whether multilingual signals can coexist without interference, and the model shows stable cross-lingual behavior across all languages. Visual rendering of multiple scripts inside the generated image itself (for example, Chinese title together with English subtitle) depends on the granularity of the image tokenizer rather than the multilingual training objective. MAGVIT-v2 is not optimized for such fine-grained typography. Instead, our focus is on semantic and compositional multilingual generation, and multi-script visual text rendering is a natural extension requiring a higher-fidelity tokenizer, which we leave for future work.

---

> ### Author Response · Authors · 2025-11-20
> **Continuation of Response to Reviewer RHTr**
>
> **Without presentation, how do the multiple languages ​​facilitate each other?** We believe the languages facilitate each other. This follows from NeoBabel’s shared multilingual tokenizer and unified embedding space, which allow semantic concepts learned from one language to reinforce those expressed in others. All six languages improve jointly through Stages 1 to 3 (Figures 7 and 11), a process that would not occur if languages were isolated or interfering.
>
> **Are there similarities between text embeddings?** Yes you are right, the text embeddings across languages are similar and aligned. The Gemma-2 tokenizer provides multi-script subword coverage, and joint training on multilingual image–text pairs enforces semantic alignment in the shared embedding space. The model’s stable performance across six scripts on m-GenEval and m-DPG, and its ability to follow equivalent prompts across languages, indicates consistent embedding structure rather than disjoint clusters.
>
> **Won’t the representations of different languages cause confusion?** We believe representation confusion does not occur. NeoBabel disambiguates languages using contextual encoding, and no language-specific parameters compete within the model. If confusion were present, some languages would degrade during training; instead, performance increases simultaneously across all languages and remains strong for English, demonstrating stable, non-interfering multilingual representations.
>
> **NeoBabel does not generate text, but enabling text generation is a straightforward extension of the current architecture.** NeoBabel is trained exclusively as an image generator. To support text generation, the model can reuse the same architecture by reordering the token streams so that visual tokens appear first and by applying an autoregressive objective only to the text tokens instead of the discrete diffusion objective. This adaptation uses the same multilingual dataset and requires no architectural changes. Text generation lies outside the scope of the present work and is left for future development. We will better stress our focus in the main paper.
>
> We hope this addresses your concerns. We appreciate your consideration and hope this motivates you to upgrade your score. Please let us know if you have any further questions or feedback.

---

> > ### Author Response · Authors · 2025-11-27
> >
> > Dear Reviewer RHTr,
> >
> > Thank you once again for taking the time and effort to review our submission. As the deadline for the discussion phase is approaching, we wanted to kindly check if you have any additional questions or concerns that we could address in our rebuttal.
> >
> > We appreciate your insights and look forward to your feedback.
> >
> > Best regards,
> >
> > The Authors

---

> ### Comment · Reviewer_RHTr · 2025-11-28
>
> I appreciate the authors’ responses, which address most of my concerns.
> I will raise my score accordingly.

---

### Official Review · Reviewer_DJPi · 2025-11-01

**Soundness:** 3
**Presentation:** 3
**Contribution:** 3
**Rating:** 6
**Confidence:** 3

**Summary:**

NEOBABEL is a novel multilingual text-to-image generation framework that supports six languages without relying on intermediate translation. Through large-scale multilingual pretraining and high-resolution instruction tuning, it achieves state-of-the-art generation performance across all supported languages, matching or exceeding the capability of larger English-only models. The model is also highly efficient, running 2.8× faster than translation-based pipelines while using ~59% less memory. The paper introduces new multilingual evaluation benchmarks and shows NEOBABEL outperforms other approaches on these, and it releases an open-source toolkit with code, model checkpoints, a curated dataset, and evaluation protocols to encourage inclusive AI research.

**Strengths:**

1. The model demonstrates strong empirical performance, achieving the highest overall score on benchmarks while using only 2B parameters. NEOBABEL matches or surpasses larger state-of-the-art models in English generation and delivers consistently high results in other languages, with especially large gains in medium- and low-resource languages (e.g. significantly outperforming baselines in Hindi and Persian).
2. The authors bolster reproducibility by open-sourcing the entire project, providing model checkpoints, a 124M multilingual image-text dataset, all code, and standardized evaluation scripts.

**Weaknesses:**

1. NEOBABEL currently supports only six languages, which, while diverse, leaves out many globally important languages. Scaling to additional languages would require non-trivial effort (e.g. adapting the tokenizer and retraining with new data).
2. The training data largely comes from translated or AI-recaptioned English captions, which might introduce biases or unnatural phrasing. Despite quality controls, relying on machine translations can risk subtle semantic shifts, and any biases in the curated dataset (or the translation models used) may be reflected in NEOBABEL’s outputs, potentially affecting fairness or cultural authenticity.
3. The evaluation focuses on the new automated benchmarks (m-GenEval and m-DPG) and examples, without reported human studies of image quality or fidelity. Important aspects like subjective image quality, user preference across languages, or the preservation of nuanced cultural concepts are not deeply evaluated, so it remains unclear how the model’s outputs are perceived by native speakers or in real-world creative applications.

**Questions:**

1. In the progressive training pipeline, Stage 2 involves fine-tuning on a large English-only subset of the data. Why did the authors choose to include an English-centric stage during pretraining, and how do they ensure this does not bias the model towards English at the expense of other languages?
2. NEOBABEL currently supports six languages with a fixed multilingual tokenizer. How easily can the approach be scaled to additional languages or scripts (e.g. adding Spanish or Arabic) – would this require retraining a new model or tokenizer, and have the authors considered the potential impact on performance when extending to a broader set of languages?

---

> ### Author Response · Authors · 2025-11-20
> **Response to Reviewer DJPi**
>
> We thank Reviewer DJPi for the constructive feedback and insightful comments.
>
> **Scaling NeoBabel to additional languages is straightforward.** Our six languages demonstrate a proof-of-concept designed to span diverse scripts and resource levels to study the model’s behavior across high-, medium-, and low-resource language regimes. We would like to clarify that extending NeoBabel to more languages is straightforward and does not require retraining the model from scratch. The model already uses the Gemma-2 multilingual tokenizer, which provides subword coverage for 100+ languages, and our unified architecture contains no language-specific components. New languages can be added through continued multilingual pretraining with additional recaptioned/translated image-text pairs, following standard practice in recent multilingual LLMs. We will clarify this in section 4 of the paper.
>
> **Image-grounded translation mitigates semantic drift.** The reviewer is right. Indeed, naïve text translation can introduce semantic drift. That is why our translation pipeline is image-grounded and generated by conditioning on the visual content rather than performing direct text-to-text translation. While grounding does not eliminate the risk of drift, guidance by the shared visual semantics reduces it to some extent. Our benchmarks also undergo human verification to minimize translation artifacts. Thus, while translation tools are used, NeoBabel’s data curation strategy systematically mitigates rather than inherits the semantic drift problem while maintaining the scalability of data curation. We will better emphasize our image-grounded translation process in the introduction.
>
> **Automated multilingual benchmarks enable scalable evaluation.** The reviewer is right. Human evaluation is valuable and prohibitively expensive, yet inconsistent when extended across multiple languages, scripts, and cultures. That is why the objective in our work is to establish automated, reproducible, and language-controlled text-to-image benchmarks that enable scalable and fair cross-lingual comparison. The proposed m-GenEval and m-DPG benchmarks, as well as the proposed applications in figures 4, 5, and 6 in the main paper and 12 to 16 in the appendix, quantify and qualify the core aspects of multilingual visual generation and provide a standardized framework that did not previously exist beyond English. A clarification will be added, and broader native speaker and cultural evaluations are noted as important lines of research for the near future.
>
> **Stage 2 enhances visual priors, not English bias.** We clarify that Stage 2 is included to strengthen visual–language alignment, not to introduce linguistic knowledge. The large-scale datasets used here (SA-1B, CC12M) offer broad visual diversity and dense grounding signals that are not available at comparable scale in multilingual form. This stage therefore improves the model’s visual priors and compositional reasoning. Crucially, Stage 2 is not trained in isolation in English. It is conducted jointly with 72M multilingual samples from m-LAION-Aesthetic, ensuring continuous cross-lingual signals throughout this phase. Because the architecture contains no language-specific components, improvements in visual grounding transfer uniformly across all supported languages rather than privileging English. This design choice is validated empirically: multilingual performance increases across every language after Stage 2 (Figures 7 and 11), demonstrating that this stage enhances generalization and does not harm non-English capability. We will better stress this in the paper.
>
> **Scaling to more languages is data-driven, not model-driven.** NeoBabel utilizes the Gemma-2 tokenizer (256K vocabulary), which already provides subword coverage for a broad range of scripts (over 100 languages) and is explicitly designed for multilingual usage, requiring no modification for additional languages such as Spanish or Arabic. Because the architecture also contains no language-specific components, extending it to additional languages primarily requires augmenting the multilingual training data and conducting continued pretraining, rather than creating a new model or tokenizer. We will better stress this in the paper.
>
> We hope this addresses your concerns. We appreciate your consideration and hope this motivates you to upgrade your score. Please let us know if you have any further questions or feedback.

---

> > ### Author Response · Authors · 2025-11-27
> >
> > Dear Reviewer DJPi,
> >
> > Thank you once again for taking the time and effort to review our submission. As the deadline for the discussion phase is approaching, we wanted to kindly check if you have any additional questions or concerns that we could address in our rebuttal.
> >
> > We appreciate your insights and look forward to your feedback.
> >
> > Best regards,
> > The Authors

---

### Author Response · Authors · 2025-11-20
**General response by the Authors**

We sincerely thank all Reviewers for their time and constructive feedback. We appreciate that our work is recognized for its strong multilingual performance and full reproducibility release (DJPi), its clear presentation and unified architecture with extended embedding space and modality-aware attention (RHTr), its contribution to non-English text-to-image generation through curated multilingual data and benchmarks (e9m4), and its motivation for moving beyond translation-first pipelines through a direct multilingual design using a single decoder-only stack with shared embeddings, image-token full attention, and special prompt tokens (bunK). We address the questions raised by the Reviewers in the respective comment sections.

---

### Meta-Review · Area_Chair_sGjp · 2026-01-09

**Summary:**

The paper proposes NeoBabel, a model for multilingual text-to-image generation supporting 6 languages: English, Chinese, Dutch, French, Hindi, Persian. It comes with curated pre-training data of 124M multilingual text-image pairs, tuning data, and benchmarks (multilingual extensions of GenEval (Ghosh et al., 2023) and DPG-Bench (Hu et al., 2024)). The paper receives 6, 4, 4, 6 ratings: Reviewer DJPi (6), Reviewer RHTr (4), Reviewer e9m4 (4), Reviewer bunK (6).

1. Soundness of the approach. Reliance on translation-based data curation: a major concern shared by 3 reviewers (Reviewer DJPi, Reviewer e9m4, Reviewer e9m4): the training data and benchmarks are translated English captions (with images as additional inputs for grounding). This is against the statement in the abstract “While existing systems rely on translation pipelines, these introduce semantic drift, computational overhead, and cultural misalignment.”

2. Quality of new benchmarks (m-GenEval and m-DPG) and data: Reviewer DJPi.

3. Scope of languages and tasks, including the limited number of languages supported (six): Reviewer DJPi.

4. Experiments. Omission of mixed languages (Reviewer RHTr), omission of fine-grained capabilities (Reviewer RHTr, e.g., can NEOBABEL generate readable text in different scripts such as Chinese, Devanagari, Persian?”). Lack of rigorous analysis: Reviewer RHTr.

5. Other minor concerns: Limited modeling contribution: Reviewer e9m4 mentions “architecture is not particularly novel”. Unproven claim on latency/memory: Reviewer bunK.

**Reviewer Concerns:**

[Not resolved] For 1 and 2, the authors argue that image-grounded translation mitigates semantic drift. However, this claim is not directly supported by any results in the paper. The authors also mention NLLB and Gemini consistently delivered the highest-quality data, but again this is not supported. Since these concerns are lingering, they hurt the soundness and significance of the approach overall. It remains unclear to which degree this benchmark is reliable.

[Borderline, Not resolved] For 3, the authors argue that extending to other languages is straightforward due to the choice of their tokenizer, but without further support.

[Borderline] For 4, the authors point to various results on the concern about mixed language evaluation, and clarify that the fine-grained capabilities hurt due to the inherent limitation of the visual tokenizer, not the approach.

**Reviewer Scores:**

Reviewer DJPi (6): Keep
Reviewer RHTr (> 4): Increase. Reviewer RHTr explicitly mentions they would raise the score.
Reviewer e9m4 (4): Keep
Reviewer bunK (6): Keep

---

### Decision · Program_Chairs · 2026-01-26

Reject